# Uncertainty on Asynchronous Time Event Prediction

**Marin Biloš**[*], **Bertrand Charpentier**[*], **Stephan Günnemann**
Technical University of Munich, Germany
`{bilos, charpent, guennemann}@in.tum.de`

## Abstract

Asynchronous event sequences are the basis of many applications throughout different industries. In this work, we tackle the task of predicting the next event (given a history), and how this prediction changes with the passage of time. Since at some time points (e.g. predictions far into the future) we might not be able to predict anything with confidence, capturing uncertainty in the predictions is crucial. We present two new architectures, WGP-LN and FD-Dir, modelling the evolution of the distribution on the probability simplex with time-dependent logistic normal and Dirichlet distributions. In both cases, the combination of RNNs with either Gaussian process or function decomposition allows to express rich temporal evolution of the distribution parameters, and naturally captures uncertainty. Experiments on class prediction, time prediction and anomaly detection demonstrate the high performances of our models on various datasets compared to other approaches.

## 1 Introduction

Discrete events, occurring irregularly over time, are a common data type generated naturally in our everyday interactions with the environment (see Fig. 2a for an illustration). Examples include messages in social networks, medical histories of patients in healthcare, and integrated information from multiple sensors in complex systems like cars. The problem we are solving in this work is: given a (past) sequence of asynchronous events, what will happen next? Answering this question enables us to predict, e.g., what action an internet user will likely perform or which part of a car might fail.

While many recurrent models for asynchronous sequences have been proposed in the past [19, 6], they are ill-suited for this task since they output a *single prediction* (e.g. the most likely next event) only. In an asynchronous setting, however, such a single prediction is not enough since the most likely event can change with the passage of time – even if no other events happen. Consider a car approaching another vehicle in front of it. Assuming nothing happens in the meantime, we can expect different events at *different times in the future*. When forecasting a short time, one expects the driver to start overtaking; after a longer time one would expect braking; in the long term, one would expect a collision. Thus, the expected behavior changes depending on the time we forecast, assuming no events occured in the meantime. Fig. 2a illustrates this schematically: having observed a square and a pentagon, it is likely to observe a square after a short time, while a circle after a longer time. Clearly, if some event occurs, e.g. braking/square, the event at the (then) observed time will be taken into account, updating the temporal prediction.

An ad-hoc solution to this problem would be to discretize time. However, if the events are near each other, a high sampling frequency is required, giving us very high computational cost. Besides, since there can be intervals without events, an artificial 'no event' class is required.

In this work, we solve these problems by directly predicting the entire evolution of the events over (continuous) time. Given a past asynchronous sequence as input, we can predict and evaluate for

---

[*]Equal contribution

*any* future timepoint what the next event will likely be (under the assumption that no other event happens in between which would lead to an update of our model). Crucially, the likelihood of the events might change and one event can be more likely than others multiple times in the future. This periodicity exists in many event sequences. For instance, given that a person is currently at home, a smart home would predict a high probability that the kitchen will be used at lunch and/or dinner time (see Fig. 1a for an illustration). We require that our model captures such multimodality.

While Fig. 1a illustrates the evolution of the categorical distribution (corresponding to the probability of a specific event class to happen), an issue still arises outside of the observed data distribution. E.g. in some time intervals we can be *certain* that two classes are *equiprobable*, having observed many similar examples. However, if the model has not seen any examples at specific time intervals during training, we do not want to give a confident prediction. Thus, we incorporate *uncertainty* in a prediction directly in our model. In places where we expect events, the confidence will be higher, and outside of these areas the uncertainty in a prediction will grow as illustrated in

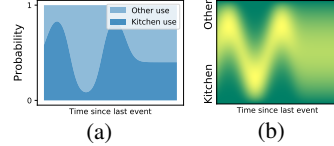

(a)  (b)

Figure 1: (a) An event can be expected multiple times in the future. (b) At some times we should be uncertain in the prediction. Yellow denotes higher probability density.

Fig. 1b. Technically, instead of modeling the evolution of a categorical distribution, we model the *evolution of a distribution on the probability simplex*. Overall, our model enables us to operate with the *asynchronous discrete* event data from the past as input to perform *continuous-time* predictions to the future incorporating the predictions' uncertainty. This is in contrast to existing works as [6, 18].

## 2   Model Description

We consider a sequence $[e_1, \ldots, e_n]$ of events $e_i = (c_i, t_i)$, where $c_i \in \{1, \ldots, C\}$ denotes the class of the $i$th event and $t_i \in \mathbb{R}$ is its time of occurrence. We assume the events arrive over time, i.e. $t_i > t_{i-1}$, and we introduce $\tau_i^* = t_i - t_{i-1}$ as the observed time gap between the $i$th and the $(i-1)$th event. The history preceding the $i$th event is denoted by $\mathcal{H}_i$. Let $S = \{\boldsymbol{p} \in [0,1]^C, \sum_c p_c = 1\}$ denote the set of probability vectors that form the $(C-1)$-dimensional simplex, and $P(\theta)$ be a family of probability distributions on this simplex parametrized by parameters $\theta$. Every sample $\boldsymbol{p} \sim P(\theta)$ corresponds to a (categorical) class distribution.

Given $e_{i-1}$ and $\mathcal{H}_{i-1}$, our goal is to model the evolution of the class probabilities, and their uncertainty, of the next event $i$ over time. Technically, we model parameters $\theta(\tau)$, leading to a distribution $P$ over the class probabilities $\boldsymbol{p}$ for all $\tau \geq 0$. Thus, we can estimate the most likely class after a time gap $\tau$ by calculating $\operatorname{argmax}_c \bar{\boldsymbol{p}}(\tau)_c$, where $\bar{\boldsymbol{p}}(\tau) := \mathbb{E}_{\boldsymbol{p}(\tau) \sim P(\theta(\tau))}[\boldsymbol{p}(\tau)]$ is the expected probability vector. Even more, since we do not consider a point estimate, we can get the amount of certainty in a prediction. For this, we estimate the probability of class $c$ being more likely than the other classes, given by $q_c(\tau) := \mathbb{E}_{\boldsymbol{p}(\tau) \sim P(\theta(\tau))}[\mathbb{1}_{\boldsymbol{p}(\tau)_c \geq \max_{c' \neq c} \boldsymbol{p}(\tau)_{c'}}]$. This tells us how certain we are that one class is the most probable (i.e. 'how often' is $c$ the argmax when sampling from $P$).

Two expressive and well-established choices for the family $P$ are the Dirichlet distribution and the logistic-normal distribution (Appendix A). Based on a common modeling idea, we present two models that exploit the specificities of these distributions: the WGP-LN (Sec. 2.1) and the FD-Dir (Sec. 2.2). We also introduce a novel loss to train these models in Sec. 2.3.

Independent of the chosen model, we have to tackle two core challenges: (1) **Expressiveness.** Since the time dependence of $\theta(\tau)$ may be of different forms, we need to capture complex behavior. (2) **Locality.** For regions out of the observed data we want to have a higher uncertainty in our predictions. Specifically for $\tau \to \infty$, i.e. far into the future, the distribution should have a high uncertainty.

### 2.1   Logistic-Normal via a Weighted Gaussian Process (WGP-LN)

We start by describing our model for the case when $P$ is the family of logistic-normal (LN) distributions. How to model a compact yet expressive evolution of the LN distribution? Our core idea is to exploit the fact that the LN distribution corresponds to a multivariate random variable whose *logits* follow a *normal distribution* – and a natural way to model the evolution of a normal distribution is a *Gaussian Process*. Given this insight, the core idea of our model is illustrated in Fig. 2: (1) we generate $M$ pseudo points based on a hidden state of an RNN whose input is a sequence, (2) we fit

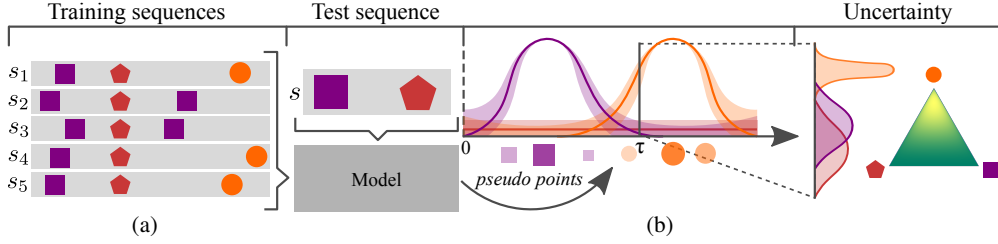

Figure 2: The model framework. (a) During training we use sequences $s_i$. (b) Given a new sequence of events $s$ the model generates pseudo points that describe $\boldsymbol{\theta}(\tau)$, i.e. the temporal evolution of the distribution on the simplex. These pseudo points are based on the data that was observed in the training examples and weighted accordingly. We also have a measure of certainty in our prediction.

a Gaussian Process to the pseudo points, thus capturing the temporal evolution, and (3) we use the learned GP for estimating the parameters $\boldsymbol{\mu}(\tau)$ and $\boldsymbol{\Sigma}(\tau)$ of the final LN distribution at any specific time $\tau$. Thus, by generating a small number of points we characterize the full distribution.

**Classic GP.** To keep the complexity low, we train one GP per class $c$. That is, our model generates $M$ points $(\tau_j^{(c)}, y_j^{(c)})$ per class $c$, where $y_j^{(c)}$ represents logits. Note that the first coordinate of each pseudo point corresponds to time, leading to the temporal evolution when fitting the GP. Essentially we perform a non-parameteric regression from the time domain to the logit space. Indeed, using a classic GP along with the pseudo points, the parameters $\theta$ of the logistic-normal distribution, $\boldsymbol{\mu}$ and $\boldsymbol{\Sigma}$, can be easily computed for any time $\tau$ in closed form:

$$\mu_c(\tau) = \boldsymbol{k}_c^T \boldsymbol{K}_c^{-1} \boldsymbol{y}_c, \; \sigma_c^2(\tau) = s_c - \boldsymbol{k}_c^T \boldsymbol{K}_c^{-1} \boldsymbol{k}_c \tag{1}$$

where $\boldsymbol{K}_c$ is the gram matrix w.r.t. the $M$ pseudo points of class $c$ based on a kernel $k$ (e.g. $k(\tau_1, \tau_2) = \exp(-\gamma^2(\tau_1 - \tau_2)^2)$). Vector $\boldsymbol{k}_c$ contains at position $j$ the value $k(\tau_j^{(c)}, \tau)$, and $\boldsymbol{y}_c$ the value $y_j^{(c)}$, and $s_c = k(\tau, \tau)$. At every time point $\tau$ the logits then follow a multivariate normal distribution with mean $\boldsymbol{\mu}(\tau)$ and covariance $\boldsymbol{\Sigma} = \text{diag}(\boldsymbol{\sigma}^2(\tau))$.

Using a GP enables us to describe complex functions. Furthermore, since a GP models uncertainty in the prediction depending on the pseudo points, uncertainty is higher in areas far away from the pseudo points. Specifically, it holds for distant future; thus, matching the idea of locality. However, uncertainty is always low around the $M$ pseudo points. Thus $M$ should be carefully picked since there is a trade-off between having high certainty at (too) many time points and the ability to capture complex behavior. Thus, in the following we present an extended version solving this problem.

**Weighted GP.** We would like to pick $M$ large enough to express rich multimodal functions and allow the model to discard unnecessary points. To do this we generate an additional weight vector $\boldsymbol{w}^{(c)} \in [0, 1]^M$ that assigns the weight $w_j^{(c)}$ to a point $\tau_j^{(c)}$. Giving a zero weight to a point should discard it, and giving 1 will return the same result as with a classic GP. To achieve this goal, we introduce a new kernel function:

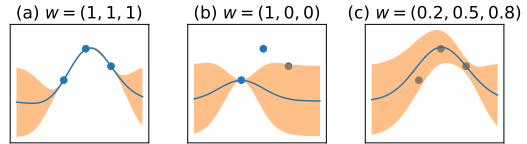

Figure 3: WGP on toy data with different weights. (a) All weights are 1 – classic GP. (b) Zero weights discard points. (c) Mixed weight assignment.

$$k'(\tau_1, \tau_2) = f(w_1, w_2)k(\tau_1, \tau_2) \tag{2}$$

where $k$ is the same as above. The function $f$ weights the kernel $k$ according to the weights for $\tau_1$ and $\tau_2$. We require $f$ to have the following properties: (1) $f$ should be a valid kernel over the weights, since then the function $k'$ is a valid kernel as well; (2) the importance of pseudo points should not increase, giving $f(w_1, w_2) \leq \min(w_1, w_2)$; this fact implies that a point with zero weight will be discarded since $f(w_1, 0) = 0$ as desired. The function $f(w_1, w_2) = \min(w_1, w_2)$ is a simple choice that fulfills these properties. In Fig. 3 we show the effect of different weights when fitting of a GP (see Appendix B for a more detailed discussion of the behavior of the $\min$ kernel).

To predict $\mu$ and $\sigma^2$ for a new time $\tau$, we can now simply apply Eq. 1 based on the new kernel $k'$, where the weight for the *query* point $\tau$ is 1.

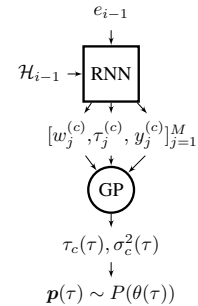

Figure 4: Model diagram

To summarize: From a hidden state $h_i = \mathrm{RNN}(e_{i-1}, \mathcal{H}_{i-1})$ we use a a neural network to generate $M$ weighted pseudo points $(w_j^{(c)}, \tau_j^{(c)}, x_j^{(c)})$ per class $c$. Fitting a Weighted GP to these points enables us to model the temporal evolution of $\mathcal{N}(\mu_c(\tau), \sigma_c^2(\tau))$ and, thus, accordingly of the logistic-Normal distribution. Fig. 4 shows an illustration of this model. Note that the cubic complexity of a GP, due to the matrix inversion, is not an issue since the number $M$ is usually small ($< 10$), while still allowing to represent rich multimodal functions. Crucially, given the loss defined in Sec. 2.3, our model is fully differentiable, enabling us efficient training.

## 2.2 Dirichlet via a Function Decomposition (FD-Dir)

Next, we consider the Dirichlet distribution to model the uncertainty in the predictions. The goal is to model the evolution of the concentrations parameters $\boldsymbol{\alpha} = (\alpha_1, \ldots, \alpha_C)^T$ of the Dirichlet over time. Since unlike to the logistic-normal, we cannot draw the connection to the GP, we propose to decompose the parameters of the Dirichlet distribution with expressive (local) functions in order to allow complex dependence on time. Since the concentration parameters $\alpha_c(\tau)$ need to be positive, we propose the following decomposition of $\alpha_c(\tau)$ in the log-space

$$\log \alpha_c(\tau) = \sum_{j=1}^{M} w_j^{(c)} \cdot \mathcal{N}(\tau | \tau_j^{(c)}, \sigma_j^{(c)}) + \nu \tag{3}$$

where the real-valued scalar $\nu$ is a constant prior on $\log \alpha_c(\tau)$ which takes over in regions where the Gaussians are close to 0.

The decomposition into a sum of Gaussians is beneficial for various reasons: (i) First note that the concentration parameter $\alpha_c$ can be viewed as the effective number of observations of class $c$. Accordingly the larger $\log \alpha$, the more certain becomes the prediction. Thus, the functions $\mathcal{N}(\tau | \tau_j^{(c)}, \sigma_j^{(c)})$ can describe time regions where we observed data and, thus, should be more certain; i.e. regions around the time $\tau_j^{(c)}$ where the 'width' is controlled by $\sigma_j^{(c)}$. (ii) Since most of the functions' mass is centered around their mean, the locality property is fulfilled. Put differently: In regions where we did not observed data (i.e. where the functions $\mathcal{N}(\tau | \tau_j^{(c)}, \sigma_j^{(c)})$ are close to 0), the value $\log \alpha_c(\tau)$ is close to the prior value $\nu$. In the experiments, we use $\nu = 0$ , thus $\alpha_c(\tau) = 1$ in the out of observed data regions; a common (uninformative) prior value for the Dirichlet parameters. Specifically for $\tau \to \infty$ the resulting predictions have a high uncertainty. (iii) Lastly, a linear combination of translated Gaussians is able to approximate a wide family of functions [4]. And similar to the weighted GP, the coefficients $w_j^{(c)}$ allow discarding unnecessary basis functions.

The basis functions parameters $(w_j^{(c)}, \tau_j^{(c)}, \sigma_j^{(c)})$ are the output of the neural network, and can also be interpreted as weighted pseudo points that determine the regression of Dirichlet parameters $\theta(\tau)$, i.e. $\alpha_c(\tau)$, over time (Fig. 2 & Fig. 4). The concentration parameters $\alpha_c(\tau)$ themselves have also a natural interpretation: they can be viewed as the rate of events after time gap $\tau$.

## 2.3 Model Training with the Distributional Uncertainty Loss

The core feature of our models is to perform predictions in the future with uncertainty. The classical cross-entropy loss, however, is not well suited to learn uncertainty on the categorical distribution since it is only based on a single (point estimate) of the class distribution. That is, the standard cross-entropy loss for the $i^{\mathrm{th}}$ event between the true categorical distribution $\boldsymbol{p}_i^*$ and the predicted (mean) categorical distribution $\overline{\boldsymbol{p}}_i$ is $\mathcal{L}_i^{\mathrm{CE}} = \mathrm{H}[\boldsymbol{p}_i^*, \overline{\boldsymbol{p}}_i(\tau_i^*)] = -\sum_c p_{ic}^* \log \overline{p}_{ic}(\tau_i^*)$. Due to the point estimate $\overline{\boldsymbol{p}}_i(\tau) = \mathbb{E}_{\boldsymbol{p}_i \sim P_i(\theta(\tau))}[\boldsymbol{p}_i]$, the uncertainty on $\boldsymbol{p}_i$ is completely neglected.

Instead, we propose the uncertainty cross-entropy which takes into account uncertainty:

$$\mathcal{L}_i^{\mathrm{UCE}} = \mathbb{E}_{\boldsymbol{p}_i \sim P_i(\theta(\tau_i^*))}[\mathrm{H}[\boldsymbol{p}_i^*, \boldsymbol{p}_i]] = -\int P_i(\theta(\tau_i^*)) \sum_c p_{ic}^* \log p_{ic} \tag{4}$$

Remark that the uncertainty cross-entropy does not use the compound distribution $\overline{\boldsymbol{p}}_i(\tau)$ but considers the expected cross-entropy. Based on Jensen's inequality, it holds: $0 \leq \mathcal{L}_i^{\mathrm{CE}} \leq \mathcal{L}_i^{\mathrm{UCE}}$. Consequently, a low value of the uncertainty cross-entropy guarantees a low value for the classic cross entropy loss, while additionally taking the variation in the class probabilities into account. A comparison between the classic cross entropy and the uncertainty cross-entropy on a simple classification task and anomaly detection in asynchronous event setting is presented in Appendix F.

In practice the true distribution $p_i^*$ is often a one hot-encoded representation of the observed class $c_i$ which simplifies the computations. During training, the models compute $P_i(\theta(\tau))$ and evaluate it at the true time of the next event $\tau_i^*$ given the past event $e_{i-1}$ and the history $\mathcal{H}_{i-1}$. The final loss for a sequence of events is simply obtained by summing up the loss for each event $\mathcal{L} = \sum_i \mathbb{E}_{p_i \sim P_i(\theta(\tau_i^*))}[\mathrm{H}[p_i^*, p_i]]$.

**Fast computation.** In order to have an efficient computation of the uncertainty cross-entropy, we propose closed-form expressions. *(1) Closed-form loss for Dirichlet.* Given that the observed class $c_i$ is one hot-encoded by $p_i^*$, the uncertain loss can be computed in closed form for the Dirichlet:

$$\mathcal{L}_i^{\mathrm{UCE}} = \mathbb{E}_{p_i(\tau_i^*) \sim \mathbf{Dir}(\alpha(\tau_i^*))}[\log p_{c_i}(\tau_i^*)] = \Psi(\alpha_{c_i}(\tau_i^*)) - \Psi(\alpha_0(\tau_i^*)) \tag{5}$$

where $\Psi$ denotes the digamma function and $\alpha_0(\tau_i^*) = \sum_c^C \alpha_c(\tau_i^*)$. *(2) Loss approximation for GP.* For WGP-LN, we approximate $\mathcal{L}_i^{\mathrm{UCE}}$ based on second order series expansion (Appendix C):

$$\mathcal{L}_i^{\mathrm{UCE}} \approx \mu_{c_i}(\tau_i^*) - \log\Big(\sum_c^C \exp(\mu_c(\tau_i^*) + \sigma_c^2(\tau_i^*)/2)\Big) + \frac{\sum_c^C (\exp(\sigma_c^2(\tau_i^*)) - 1)\exp(2\mu_c(\tau_i^*) + \sigma_c^2(\tau_i^*))}{2\Big(\sum_c^C \exp(\mu_c(\tau_i^*) + \sigma_c^2(\tau_i^*)/2)\Big)^2} \tag{6}$$

Note that we can now fully backpropagate through our loss (and through the models as well), enabling to train our methods efficiently with automatic differentiation frameworks and, e.g., gradient descent.

**Regularization.** While the above loss much better incorporates uncertainty, it is still possible to generate pseudo points with high weight values outside of the observed data regime giving us predictions with high confidence. To eliminate this behaviour we introduce a regularization term $r_c$:

$$r_c = \alpha \underbrace{\int_0^T (\mu_c(\tau))^2 \, d\tau}_{\text{Pushes mean to 0}} + \beta \underbrace{\int_0^T (\nu - \sigma_c^2(\tau))^2 \, d\tau}_{\text{Pushes variance to } \nu} \tag{7}$$

For the WGP-LN, $\mu_c(\tau)$ and $\sigma_c(\tau)$ correspond to the mean and the variance of the class logits which are pushed to prior values of $0$ and $\nu$. For the FD-Dir, $\mu_c(\tau)$ and $\sigma_c(\tau)$ correspond to the mean and the variance of the class probabilities where the regularizer on the mean can actually be neglected because of the prior $\nu$ introduced in the function decomposition (Eq. 3). In experiments, $\nu$ is set to $1$ for WGP-LN and $\frac{C-1}{C^2(C+1)}$ for FD-Dir which is the variance of the classic Dirichlet prior with concentration parameters equal to $1$. For both models, this regularizer forces high uncertainty on the interval $(0, T)$. In practice, the integrals can be estimated with Monte-Carlo sampling whereas $\alpha$ and $\beta$ are hyperparameters which are tuned on a validation set.

In [16], to train models capable of uncertain predictions, another dataset or a generative models to access out of observed distribution samples is required. In contrast, our regularizer suggests a simple way to consider out of distribution data which does not require another model or dataset.

## 3  Point Process Framework

Our models FD-Dir and WGP-LN predict $P(\theta(\tau))$, enabling to evaluate, e.g., $\overline{p}$ after a specific time gap $\tau$. This corresponds to a conditional distribution $q(c|\tau) := \overline{p}_c(\tau)$ over the classes. In this section, we introduce a *point process* framework to generalize FD-Dir to also predict the time distribution $q(\tau)$. This enables us to predict, e.g., the most likely time the next event is expected or to evaluate the joint distribution $q(c|\tau) \cdot q(\tau)$. We call the model FD-Dir-PP.

We modify the model so that each class $c$ is modelled using an inhomogeneous Poisson point process with positive locally integrable intensity function $\lambda_c(\tau)$. Instead of generating parameters $\theta(\tau) = (\alpha_1(\tau), ..., \alpha_C(\tau))$ by function decomposition, FD-Dir-PP generates intensity parameters over time: $\log \lambda_c(\tau) = \sum_{j=1}^M w_j^{(c)} \mathcal{N}(\tau|\tau_j^{(c)}, \sigma_j^{(c)}) + \nu$. The main advantage of such general decomposition is its potential to describe complex multimodal intensity functions contrary to other models like RMTPP [6] (Appendix D). Since the concentration parameter $\alpha_c(\tau)$ and the intensity parameter $\lambda_c(\tau)$ both relate to the number of events of class $c$ around time $\tau$, it is natural to convert one to the other.

Given this $C$-multivariate point process, the probability of the next class given time and the probability of the next event time are $q(c|\tau) = \frac{\lambda_c(\tau)}{\lambda_0(\tau)}$ and $q(\tau) = \lambda_0(\tau)e^{-\int_0^\tau \lambda_0(s)ds}$ where $\lambda_0(\tau) = \sum_{c=1}^C \lambda_c(\tau)$.

Since the classes are now modelled via a point proc., the log-likelihood of the event $e_i = (c_i, \tau_i^*)$ is:

$$\log q(c_i, \tau_i^*) = \log q(c_i | \tau_i^*) + \log q(\tau_i^*) = \underbrace{\log \frac{\lambda_{c_i}(\tau_i^*)}{\lambda_0(\tau_i^*)}}_{(i)} + \underbrace{\log \lambda_0(\tau_i^*)}_{(ii)} - \underbrace{\int_0^{\tau_i^*} \lambda_0(t) dt}_{(iii)} \qquad (8)$$

The terms (ii) and (iii) act like a regularizer on the intensities by penalizing large cumulative intensity $\lambda_0(\tau)$ on the time interval $[t_{i-1}, t_i]$ where no events occurred. The term (i) is the standard cross-entropy loss at time $\tau_i$. Or equivalently, by modeling the distribution $\mathbf{Dir}(\lambda_1(\tau), .., \lambda_C(\tau))$, we see that term (i) is equal to $\mathcal{L}_i^{\text{CE}}$ (see Section 2.3). Using this insight, we obtain our final FD-Dir-PP model: We achieve uncertainty on the class prediction by modeling $\lambda_c(\tau)$ as concentration parameters of a Dirichlet distribution and train the model with the loss of Eq. 8 replacing term (i) by $\mathcal{L}_i^{\text{UCE}}$. As it becomes apparent FD-Dir-PP differs from FD-Dir only in the regularization of the loss function, enabling it to be interpreted as a point process.

## 4  Related Work

Predictions based on discrete sequences of events regardless of time can be modelled by Markov Models [2] or RNNs, usually with its more advanced variants like LSTMs [11] and GRUs [5]. To exploit the time information some models [15, 19] additionally take time as an input but still output a single prediction for the entire future. In contrast, temporal point process framework defines the intensity function that describes the rate of events occuring over time.

RMTPP [6] uses an RNN to encode the event history into a vector that defines an exponential intensity function. Hence, it is able to capture complex past dependencies and model distributions resulting from simple point processes, such as Hawkes [10] or self-correcting [12], but not e.g. multimodal distributions. On the other hand, Neural Hawkes Process [18] uses continuous-time LSTM which allows specifying more complex intensity functions. Now the likelihood evaluation is not in closed-form anymore, but requires Monte Carlo integration. However, these approaches, unlike our models, do not provide any uncertainty in the predictions. In addition, WGP-LN and FD-Dir can be extended with a point process framework while having the expressive power to represent complex time evolutions.

Uncertainty in machine learning has shown a great interest [9, 8, 14]. For example, uncertainty can be imposed by introducing distributions over the weights [3, 17, 20]. Simpler approaches introduce uncertainty directly on the class prediction by using Dirichlet distribution independent of time [16, 21]. In contrast, the FD-Dir model models complex temporal evolution of Dirichlet distribution via function decomposition which can be adapted to have a point process interpretation.

Other methods introduce uncertainty time series prediction by learning state space model with Gaussian processes [7, 23]. Alternatively, RNN architecture has been used to model the probability density function over time [25]. Compared to these models, the WGP-LN model uses both Gaussian processes and RNN to model uncertainty and time. Our models are based on pseudo points. Pseudo points in a GP have been used to reduce the computational complexity [22]. Our goal is not to speed up the computation, since we control the number of points that are generated, but to give them different importance. In [24] a weighted GP has been considered by rescaling points; in contrast, our model uses a custom kernel to discard (pseudo) points.

## 5  Experiments

We evaluate our models on large-scale synthetic and real world data. We compare to neural point process models: **RMTPP** [6] and **Neural hawkes process** [18]. Additionally, we use various RNN models with the knowledge of the time of the next event. We measure the accuracy of class prediction, accuracy of time prediction, and evaluate on an anomaly detection task to show prediction uncertainty.

We split the data into train, validation and test set (60%–20%–20%) and tune all models on a validation set using grid search over learning rate, hidden state dimension and $L_2$ regularization. After running models on all datasets 5 times we report mean and standard deviation of test set accuracy. Details

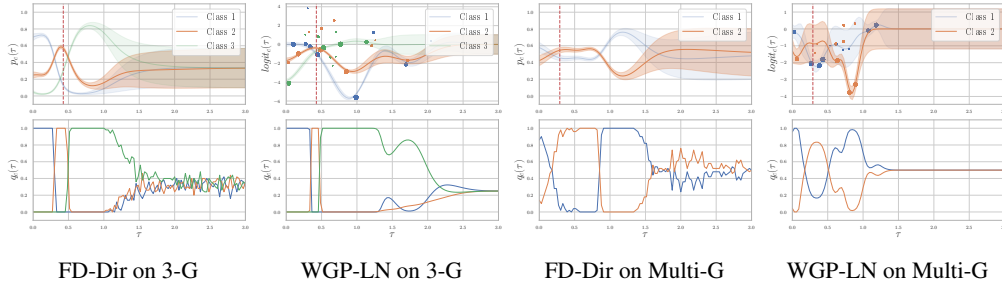

| FD-Dir on 3-G | WGP-LN on 3-G | FD-Dir on Multi-G | WGP-LN on Multi-G |

Figure 5: Visualization of the prediction evolution. The red line indicates the true time of the next event for an example sequence. Here, both models predict the orange class, which is correct, and capture the variation of the class distributions over time. Generated points from WGP-LN are plotted with the size corresponding to the weight. For predictions in the far future, both models given high uncertainty.

on model selection can be found in Appendix H.1. The code and further supplementary material is available online.[2]

We use the following data (more details in Appendix G): (1) **Graph.** We generate data from a directed Erdős–Rényi graph where nodes represent the states and edges the weighted transitions between them. The time it takes to cross one edge is modelled with one normal distribution per edge. By randomly walking along this graph we created 10K asynchronous events with 10 unique classes. (2) **Stack Exchange.**[3] Sequences contain rewards as events that users get for participation on a question answering website. After preprocessing according to [6] we have 40 classes and over 480K events spread over 2 years of activity of around 6700 users. The goal is to predict the next reward a user will receive. (3) **Smart Home** [1].[4] We use a recorded sequence from a smart house with 14 classes and over 1000 events. Events correspond to the usage of different appliances. The next event will depend on the time of the day, history of usage and other appliances. (4) **Car Indicators.** We obtained a sequence of events from car's indicators that has around 4000 events with 12 unique classes. The sequence is highly asynchronous, with $\tau$ ranging from milliseconds to minutes.

**Visualization.** To analyze the behaviour of the models, we propose visualizations of the evolutions of the parameters predicted by FD-Dir and WGP-LN.

*Set-up:* We use two toy datasets where the probability of an event depends only on time. The first one (**3-G**) has three classes occuring at three distinct times. It represents the events in the Fig. 13a. The second one (**Multi-G**) consists of two classes where one of them has two modes and corresponds to the Fig. 1a. We use these datasets to showcase the importance of time when predicting the next event. In Fig. 5, the four top plots show the evolution of the categorical distribution for the FD-Dir and the logits for the WGP-LN with 10 points each. The four bottom plots describe the certainty of the models on the probability prediction by plotting the probability $q_c(\tau)$ that the probability of class $c$ is higher than others, as introduced in Sec. 2. Additionally, the evolution of the dirichlet distribution over the probability simplex is presented in Appendix E.

*Results.* Both models learn meaningful evolutions of the distribution on the simplex. For the 3-G data, we can distinguish four areas: the first three correspond to the three classes; after that the prediction is uncertain. The Multi-G data shows that both models are able to approximate multimodal evolutions.

**Class prediction accuracy.** The aim of this experiment is to assess whether our models can correctly predict the class of the next event, given the time at which it occurs. For this purpose, we compare our models against Hawkes and RMTPP and evelute prediction accuracy on the test set.

*Results.* We can see (Fig. 6) that our models consistently outperform the other methods on all datasets. Results of the other baselines can be found in Appendix H.2.

**Time-Error evaluation.** Next, we aim to assess the quality of the time intervals at which we have confidence in one class. Even though WGP-LN and the FD-Dir do not model a distribution on

[2]https://www.daml.in.tum.de/uncertainty-event-prediction

[3]https://archive.org/details/stackexchange

[4]https://sites.google.com/site/tim0306/datasets

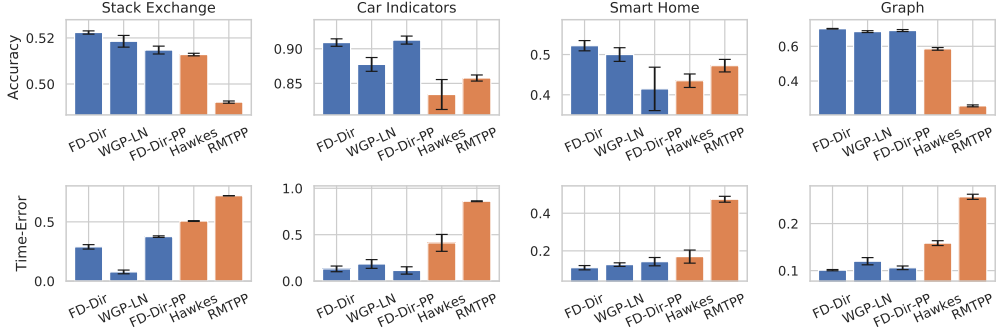

Figure 6: Class accuracy (top; higher is better) and Time-Error (bottom; lower is better).

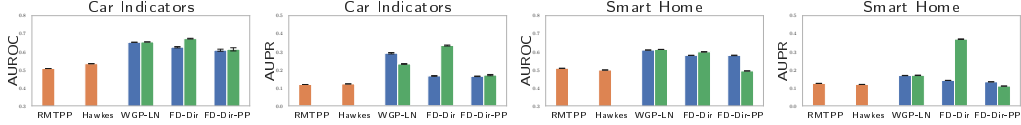

Figure 7: AUROC and APR comparison across dataset on anomaly detection. The orange and blue bars use categorical uncertainty score whereas the green bars use distributional uncertainty.

time, they still have intervals at which we are certain in a class prediction, making the conditional probability a good indicator of the time occurrence of the event.

*Set-up.* While models predicting a *single* time $\hat{\tau}_i$ for the next event often use the MSE score $\frac{1}{n}\sum_{i=1}^{n}(\hat{\tau}_i - \tau_i^*)^2$, in our case the MSE is not suitable since one event can occur at multiple time points. In the conventional least-squares approach, the mean of the true distribution is an optimal prediction; however, here it is almost always wrong. Therefore, we use another metric which is better suited for multimodal distributions. Assume that a model returns a score function $g_i^{(c)}(\tau)$ for each class regarding the next event $i$, where a large value means the class $c$ is likely to occur at time $\tau$. We define Time-Error $= \frac{1}{n}\sum_{i=1}^{n}\int \mathbb{1}_{g_i^{(c)}(\tau) \geq g_i^{(c)}(\tau_i^*)} d\tau$. The Time-Error computes the size of the time intervals where the predicted score is larger than the score of the observed time $\tau_i^*$. Hence, a performant model would achieve a low Time-Error if its score function $g_i^{(c)}(\tau)$ is high at time $\tau^*$. As the score function in our models, we use the corresponding class probability $\bar{p}_{ic}(\tau)$.

*Results.* We can see that our models clearly obtain the best results on all datasets. The point process version of FD-Dir does not improve the performance. Thus, taking also into account the class prediction performance, we recommend to use our other two models. In Appendix H.3 we compare FD-Dir-PP with other neural point process models on time prediction using the MSE score and achieve similar results.

**Anomaly detection & Uncertainty.** The goal of this experiment is twofold: (1) it assesses the ability of the models to detect anomalies in asynchronous sequences, (2) it evaluates the quality of the predicted uncertainty on the categorical distribution. For this, we use a similar set-up as [16].

*Set-up:* The experiments consist in introducing anomalies in datasets by changing the occurrence time of 10% of the events (at random after the time transformation described in appendix G). Hence, the anomalies form out-of-distribution data, whereas unchanged events represent in-distribution data. The performance of the anomaly detection is assessed using Area Under Receiver Operating Characteristic (AUROC) and Area Under Precision-Recall (AUPR). We use two approaches: (i) We consider the *categorical uncertainty* on $\bar{p}(\tau)$, i.e., to detect anomalies we use the predicted probability of the true event as the anomaly score. (ii) We use the *distribution uncertainty* at the observed occurrence time provided by our models. For WGP-LN, we can evaluate $q_c(\tau)$ directly (difference of two normal distributions). For FD-Dir, this probability does not have a closed-form solution so instead, we use the concentration parameters which are also indicators of out-of-distribution events. For all scores, i.e $\bar{p}(\tau)_c$, $q_c(\tau)$ and $\alpha_c(\tau)$, a low value indicates a potential anomaly around time $\tau$.

*Results.* As seen in Fig. 5, the FD-Dir and the WGP-LN have particularly good performance. We observe that the FD-Dir gives better results especially with distributional uncertainty. This might be due to the power of the concentration parameters that can be viewed as number of similar events around a given time.

# 6  Conclusion

We proposed two new methods to predict the evolution of the probability of the next event in asynchronous sequences, including the distributions' uncertainty. Both methods follow a common framework consisting in generating pseudo points able to describe rich multimodal time-dependent parameters for the distribution over the probability simplex. The complex evolution is captured via a Gaussian Process or a function decomposition, respectively; still enabling easy training. We also provided an extension and interpretation within a point process framework. In the experiments, WGP-LN and FD-Dir have clearly outperformed state-of-the-art models based on point processes; for event and time prediction as well as for anomaly detection.

**Acknowledgement**

This research was supported by the German Federal Ministry of Education and Research (BMBF), grant no. 01IS18036B, and by the BMW AG. The authors would like to thank Bernhard Schlegel for helpful discussion and comments. The authors of this work take full responsibilities for its content.

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
