[Supplementary Material · nips_2019_supp.pdf]

# Supplementary Materials: Uncertainty on Asynchronous Time Event Prediction

## A  Distributions

For reference, we give here the definition of the Dirichlet and Logistic-normal distribution.

### A.1  Dirichlet distribution

The Dirichlet distribution with concentration parameters $\boldsymbol{\alpha} = (\alpha_1, \ldots, \alpha_K)$, where $\alpha_i > 0$, has the probability density function:

$$f(\boldsymbol{x}; \boldsymbol{\alpha}) = \frac{\prod_{i=1}^{K} \Gamma(\alpha_i)}{\Gamma\left(\sum_{i=1}^{K} \alpha_i\right)} \prod_{i=1}^{K} x_i^{\alpha_i - 1} \tag{A.1}$$

where $\Gamma$ is a gamma function:

$$\Gamma(\alpha) = \int_0^\infty \alpha^{z-1} e^{-\alpha} dz$$

### A.2  Logistic-normal distribution (LN)

The logistic normal distribution is a generalization of the logit-normal distribution for the multidimensional case. If $\boldsymbol{y} \in \mathbb{R}^C$ follows a normal distribution, $\boldsymbol{y} \sim \mathcal{N}(\boldsymbol{\mu}, \boldsymbol{\Sigma})$, then

$$\boldsymbol{x} = \left[ \frac{e^{y_1}}{\sum_{i=1}^{C} e^{y_i}}, \ldots, \frac{e^{y_C}}{\sum_{i=1}^{C} e^{y_i}} \right]$$

follows a logistic-normal distribution.

## B  Behavior of the min kernel

The desired behavior of the min kernel function can easily be illustrated by considering the gram matrix $\boldsymbol{K}$ and vector $\boldsymbol{k}$, which are required to estimate $\mu$ and $\sigma^2$ for a new time point $\tau$. W.l.o.g. consider $M$ pseudo points $\tau_1, \ldots, \tau_M$ such that $w_1 < \cdots < w_M$. Since the new query point is observed we assign it weight 1. It follows:

$$\boldsymbol{k} = \begin{bmatrix} w_1 \\ w_2 \\ \vdots \\ w_M \end{bmatrix} \odot \begin{bmatrix} k(\tau_1, \tau) \\ k(\tau_2, \tau) \\ \vdots \\ k(\tau_M, \tau) \end{bmatrix}, \quad \boldsymbol{K} = \begin{bmatrix} w_1 & w_1 & \ldots & w_1 \\ w_1 & w_2 & \ldots & w_2 \\ \vdots & \vdots & \ddots & \vdots \\ w_1 & w_2 & \ldots & w_M \end{bmatrix} \odot \begin{bmatrix} k(\tau_1, \tau_1) & \ldots & k(\tau_1, \tau_M) \\ k(\tau_2, \tau_1) & \ldots & k(\tau_2, \tau_M) \\ \vdots & \ddots & \vdots \\ k(\tau_M, \tau_1) & \ldots & k(\tau_M, \tau_M) \end{bmatrix}$$
$$\tag{B.2}$$

Assuming $w_1 = 0$ returns $\boldsymbol{k}$ without the first row and $\boldsymbol{K}$ without the first row and column. Plugging them back into equation 1 we can see that the point $\tau_1$ is discarded, as desired. In practice, the weights have values from interval $[0, 1]$ which in turn gives us the ability to *softly discard* points. This is shown in Fig. 3 we can see that the mean line does not have to cross through the points with weights $< 1$ and the variance can remain higher around them.

## C  Computation of the approximation for the uncertainty cross-entropy of WGP-LN

Given true categorical distribution $\boldsymbol{p}_i^*$, and predicted $\boldsymbol{p}_i(\tau)$, the uncertainty cross-entropy can be calculated as in Eq. 4. For the WGP-LN model $\boldsymbol{p}_i(\tau) = \text{softmax}(\boldsymbol{z}_i(\tau))$, where $\boldsymbol{z}_i(\tau)$ are logits that come from a Gaussian process and follow a normal distribution $\mathcal{N}(\boldsymbol{\mu}_i(\tau), \boldsymbol{\Sigma}_i(\tau))$,. Therefore, $\exp(\boldsymbol{z}_i(\tau))$ follows a log-normal distribution. We will use this to derive an approximation of the loss.

From now on, we omit $\tau$ from the equations. Mean and variance for $\sum_c^C \exp(\boldsymbol{z}_{c_i})$ are then:

$$\mathbb{E}\left[\sum_c^C \exp(\boldsymbol{z}_{c_i})\right] = \sum_c^C \exp(\boldsymbol{\mu}_{c_i} + \boldsymbol{\sigma}_{c_i}^2/2)$$

$$\mathbf{Var}\left[\sum_{c_i}^C \exp(\boldsymbol{z}_{c_i})\right] = \sum_{c_i}^C (\exp(\sigma_{c_i}^2) - 1) \exp(2\boldsymbol{\mu}_{c_i} + \boldsymbol{\sigma}_{c_i}^2)$$

(C.3)

The expectation of the cross entropy loss given that logits are following a normal distribution is

$$\mathcal{L}_i^{\text{UCE}} = \mathbb{E}[\mathcal{L}_i^{\text{CE}}] = \mathbb{E}[\log(\exp(\boldsymbol{z}_{c_i}))] - \mathbb{E}\left[\log\left(\sum_c^C \exp(\boldsymbol{z}_{c_i})\right)\right]$$

(C.4)

In general, given a random variable $x$, we can approximate expectation of $\log x$ by performing a second order Taylor expansion around the mean $\mu$:

$$\mathbb{E}[\log x] \approx \mathbb{E}\left[\log \mu + \underbrace{\frac{(\log \mu)'}{1!}(x - \mu)}_{\mathbb{E}[x-\mu]=0} + \frac{(\log \mu)''}{2!}(x - \mu)^2\right]$$

$$\approx \mathbb{E}[\log \mu] - \frac{\mathbf{Var}[x]}{2\mu^2}$$

(C.5)

Using C.5 together with C.3 and plugging into C.4 we get a closed-form solution for the loss for event $i$:

$$\mathcal{L}_i^{\text{UCE}} \approx \mu_{c_i}(\tau_i^*) - \log\left(\sum_c^C \exp(\mu_c(\tau_i^*) + \sigma_c^2(\tau_i^*)/2)\right) + \frac{\sum_c^C (\exp(\sigma_c^2(\tau_i^*)) - 1) \exp(2\mu_c(\tau_i^*) + \sigma_c^2(\tau_i^*))}{2\left(\sum_c^C \exp(\mu_c(\tau_i^*) + \sigma_c^2(\tau_i^*)/2)\right)^2}$$

(C.6)

## D    Non Expressiveness of RMTPP intensities

The intensity function has the following form in the RMTPP model [5]:

$$\log \lambda_0(t) = \boldsymbol{v}^T \cdot \boldsymbol{h}_i + w(t - t_i) + b$$

(D.7)

The variables $\boldsymbol{v}$, $w$ and $b$ are learned parameters and $\boldsymbol{h}_i$ is given by the hidden state of an RNN. The only dependence on $t$ is $(t - t_i)$. RMTPP is then limited to monotonic intensity functions with respect to time.

## E    Dirichlet Evolution

Our goal is to model the evolution of a distribution on a probability simplex. Fig. 1b shows this for two classes. In general, we can do the same for multiple classes. Fig. 8 shows an example of the Dirichlet distribution for three classes, and how it changes over time. This evolution is the output of the FD-Dir model trained on the 3-G dataset, created to simulate the car example from Sec. 1 (see also Fig. 13a in Appendix G). The three classes: *overtaking*, *breaking* and *collision* occur independently of each other at three different times. The represent the corners of the triangle in Fig. 8.

We can distinguish three cases: (a) at first we are certain that the most likely class is *overtaking*; (b) as time passes, the most likely class becomes *breaking*, (c) and finally *collision*. After that, we are in the area where we have not seen any data and do not have a confident prediction (d).

(a) $\tau = 0$       (b) $\tau = 0.5$       (c) $\tau = 1$.       (d) $\tau = 2$.

Figure 8: Dirichlet distribution at different time for the 3-G dataset with $\sigma = 1$.

# F  Comparison of the classical cross-entropy and the uncertainty cross-entropy

## F.1  Simple classification task

In this section, we do not consider temporal data. The goal of this experiment is to show the benefit of the uncertainty cross-entropy compare with the classical cross-entropy loss on a simple classification task. As a consequence, we do not consider RNN in this section. We use a simple two layers neural network to predict the concentration parameters of a Dirichlet distribution from the input vector.

*Set-up.* The set-up is similar to [15] and consists of two datasets of 1500 instances divided in three equidistant 2-D Gaussians. One dataset contains non-overlapping classes (**NOG**) whereas the other contains overlapping classes (**OG**). Given one input $x_i$, we train simple two layers neural networks to predict the concentration parameters of a Dirichlet distribution $\mathbf{Dir}(\alpha_1(x_i), \alpha_2(x_i), \alpha_3(x_i))$ which model the uncertainty on the categorical distribution $p(x_i)$. On each dataset, we train two neural networks. One neural network is trained with the classic cross-entropy loss $\mathcal{L}^{\mathrm{CE}}$ which uses only the mean prediction $\bar{p}(x_i)$. The second neural network is trained with the uncertainty cross-entropy loss plus a simple $\alpha$-regularizer:

$$\mathcal{L}^{\mathrm{UCE}} + \left| \alpha_0(x_i) - \sum_j \mathbb{1}_{x_j \in N_w(x_i)} \right| \tag{F.8}$$

where $x_i$ is the input 2-D vector and $N_w(x_i) = \{x', ||x' - x_i||_2^2 < w\}$ is its euclidean neighbourhood of size $w$. We set $w = 10^{-5}$ for the non-overlapping Gaussians and $w = 10^{-2}$ for the overlapping Gaussians.

*Results.* The categorical entropy $-\sum_c p_c(x_i) \log p_c(x_i)$ is a good indicator to know how certain is the categorical distribution $p(x_i)$ at point $x_i$. A high entropy meaning that the categorical distribution is uncertain. For non overlapping Gaussians (Fig. 9a and 9b), we remark that both losses learn uncertain categorical distribution only on thin borders. However, for overlapping Gaussians (See Fig. 9c and 9d),the uncertainty cross-entropy loss learns more uncertain categorical distributions because of the thicker borders.

Other interesting results are the concentration parameters learned by the two models (Fig. 10, Fig. 11). The classic cross-entropy loss learns very high value for $\alpha_1(x_i), \alpha_2(x_i), \alpha_3(x_i)$ which does match with the true distribution of the data. In contrast, the uncertainty cross-entropy learn meaningful alpha values for both datasets (delimiting the in-distribution areas for $\alpha_0$ and centred around the classes for the others).

(a) NOG - CE - Cat. Ent.  (b) NOG - UCE - Cat. Ent.  (c) OG - CE - Cat. Ent.  (d) OG - UCE - Cat. Ent.

Figure 9: The Figures 9a and 9b plot the entropy of the categorical distribution learned on a classification task with three non-overlapping Gaussians. They show categorical entropy learned with the classic cross-entropy and learned with the uncertainty cross-entropy. The Figures 9c and 9d plot the entropy of the categorical distribution learned on a classification task with three overlapping Gaussians. They show categorical entropy learned with the classic cross-entropy and learned with the uncertainty cross-entropy.

(a) CE - $\alpha_0$    (b) CE - $\alpha_1$    (c) CE - $\alpha_2$    (d) CE - $\alpha_3$

(e) UCE - $\alpha_0$    (f) UCE - $\alpha_1$    (g) UCE - $\alpha_2$    (h) UCE - $\alpha_3$

Figure 10: Concentration parameters of the Dirichlet distribution on a classification task with three non-overlapping Gaussians. The figures 10a, 10b, 10c, 10d are $\alpha_0$, $\alpha_1$, $\alpha_2$, $\alpha_3$ learned with the classic cross-entropy. The figures 10a, 10b, 10c, 10d are $\alpha_0$, $\alpha_1$, $\alpha_2$, $\alpha_3$ learned with the uncertainty cross-entropy.

(a) CE - $\alpha_0$    (b) CE - $\alpha_1$    (c) CE - $\alpha_2$    (d) CE - $\alpha_3$

(e) UCE - $\alpha_0$    (f) UCE - $\alpha_1$    (g) UCE - $\alpha_2$    (h) UCE - $\alpha_3$

Figure 11: Concentration parameters of the Dirichlet distribution on a classification task with three non-overlapping Gaussians. The figures 11a, 11b, 10c, 11d are $\alpha_0$, $\alpha_1$, $\alpha_2$, $\alpha_3$ learned with the classic cross-entropy. The figures 11a, 11b, 10c, 11d are $\alpha_0$, $\alpha_1$, $\alpha_2$, $\alpha_3$ learned with the uncertainty cross-entropy.

## F.2 Asynchronous Event Prediction

In this section, we consider temporal data. The goal of this experiment is again to show the benefit of the uncertainty cross-entropy compared to the classical cross-entropy in the case of asynchronous event prediction.

*Set-up.* For this purpose, we use the same set-up describe in the experiment Anomaly detection & Uncertainty. We trained the model FD-Dir with three different type of losses: (1) The classical cross-entropy (CE), (2) The classical cross-entropy with regularization described in section 2.3 (CE + reg) and (3) The classical uncertainty cross-entropy with regularization described in section 2.3 (UCE + reg).

Figure 12: Loss comparison in anomaly detection

*Results.* The results are shown in Fig. 12. The loss UCE + reg consistently improves the anomaly detection based on the distribution uncertainty.

## G Datasets

In this section we describe the datasets in more detail. The time gap between two events $\tau_i^* = t_i - t_{i-1}$ is first log-transformed before applying min-max normalization: $\hat{\tau}_i^* = \frac{\tau_i' - \min(\tau_i^{*'})}{(\max(\tau_i^{*'}) - \min(\tau_i^{*'}))}$ with $\tau_i^{*'} = \log(\tau_i^* + \epsilon), \epsilon > 0$.

**3-G.** We use $C = 3$ and draw from a normal distribution $P(\tau|c_i) = \mathcal{N}(i+1, 1.)$. This dataset tries to imitate the setting from Fig. 13a as explained in 1. We generate 1000 events. Probability density is shown in figure 13b. Models that are not taking time into account cannot solve this problem. Below is the code. We create the **Multi-G** dataset similarly.

(a) Car example explained in section 1 where probabilities of events to occur change over time

(b) Probability density of events in K-Gaussians dataset. We can see that classes are independent of history.

```
def generate():
    data = np.zeros((1000, 2))
    for i in range(1000):
        i_class = np.random.choice(3, 1)[0]
        time = np.random.normal(i_class + 1, 1.)
        while time <= 0:
            time = np.random.normal(i_class + 1, 1.)
        data[i, 0] = i_class
        data[i, 1] = time
    return data
```

**Car Indicators.**  A sequence contains signals from a single car during one ride. We remove signals that are perfectly correlated giving 6 unique classes in the end. Top 3 classes make up 33%, 32%, and 16% of a total respectively. From figure 14 we can see that the setting is again asynchronous.

Figure 14: Probability density of events in Car Indicators dataset for 2 selected classes. Time is log-transformed.

**Graph.**  We generate graph $G$ with 10 nodes and 48 edges between them. We assign variables $\mu$ and $\sigma$ to each transition (edge) between events (nodes). The time it takes to make a transition between nodes $i$ and $j$ is drawn from normal distribution $\mathcal{N}(\mu_{ij}, \sigma_{ij}^2)$. By performing a random walk on the graph we create 10 thousand events. This dataset is similar to K-Gaussians with the difference that a model needs to learn the relationship between events together with the time dependency. Parts of the trace are shown in figure 15.

Figure 15: Trace of events for random graph. Different colors represent different classes and width of a single column represents the time that passed.

# H  Details of experiments

We test our models (**WGP-LN**, **FD-Dir** and **DPP**) against neural point process models (**RMTPP** and **Hawkes**) and simple baselines (**RNN** and **LSTM** – getting only history as an input, **F-RNN** and **F-LSTM** – having also the real time of the next event as an additional input; thus, they get a strong advantage!). We test on real world (**Stack Exchange**, **Car Indicators** and **Smart Home**) and synthetic datasets (**Graph**). We show that our models consistently outperform all the other models when evaluated with class prediction accuracy and Time-Error.

## H.1  Model selection

We apply the same tuning technique to all models. We split all datasets into train–validation–test sets ($60\% - 20\% - 20\%$), use the validation set to select a model and the test set to get final scores. For Stack Exchange dataset we split on users. In all other datasets we split the trace based on time. We search over dimension of a hidden state $\{32, 64, 128, 256\}$, batch size $\{16, 32, 64\}$ and $L_2$ regularization parameter $\{0, 10^{-3}, 10^{-2}, 10^{-1}\}$. We use the same learning rate 0.001 for all models and an Adam optimizer [12], run each of them 5 times for maximum of 100 epochs with early stopping after 5 consecutive epochs without improvement in the validation loss. For the number of points $M$ we pick 3 for WGP-LN and 20 for FD-Dir. WGP-LN and FD-Dir have additional regularization (Eq. 7) with hyperparameters $\alpha$ and $\beta$. For both models we choose $\alpha = \beta = 10^{-3}$. Model with the highest mean accuracy on the validation set is selected. We use GRU cell [4] for both of our models. We trained all models on GPUs (1TB SSD).

Table 1: Class accuracy comparison for all models on all datasets

|  | Car Indicators | Graph | Smart Home | Stack Exchange |
|---|---|---|---|---|
| FD-Dir | $0.909 \pm 0.005$ | $\mathbf{0.701 \pm 0.002}$ | $\mathbf{0.522 \pm 0.013}$ | $\mathbf{0.522 \pm 0.001}$ |
| Dir-PP | $\mathbf{0.912 \pm 0.006}$ | $0.691 \pm 0.006$ | $0.415 \pm 0.054$ | $0.515 \pm 0.002$ |
| WGP-LN | $0.877 \pm 0.010$ | $0.685 \pm 0.005$ | $0.500 \pm 0.017$ | $0.519 \pm 0.003$ |
| Hawkes | $0.834 \pm 0.022$ | $0.585 \pm 0.008$ | $0.435 \pm 0.017$ | $0.513 \pm 0.001$ |
| RMTPP | $0.858 \pm 0.004$ | $0.257 \pm 0.005$ | $0.472 \pm 0.016$ | $0.492 \pm 0.000$ |
| F-LSTM | $0.855 \pm 0.006$ | $0.657 \pm 0.002$ | $0.411 \pm 0.029$ | - |
| F-RNN | $0.849 \pm 0.013$ | $0.615 \pm 0.011$ | $0.472 \pm 0.035$ | - |
| LSTM | $0.858 \pm 0.010$ | $0.251 \pm 0.008$ | $0.375 \pm 0.026$ | - |
| RNN | $0.838 \pm 0.016$ | $0.258 \pm 0.008$ | $0.437 \pm 0.017$ | - |

Table 2: Time-Error comparison for all models on all datasets

|  | Car Indicators | Graph | Smart Home | Stack Exchange |
|---|---|---|---|---|
| FD-Dir | $\mathbf{0.115 \pm 0.040}$ | $\mathbf{0.101 \pm 0.001}$ | $\mathbf{0.111 \pm 0.011}$ | $0.289 \pm 0.019$ |
| WGP-LN | $0.184 \pm 0.047$ | $0.120 \pm 0.008$ | $0.127 \pm 0.010$ | $\mathbf{0.077 \pm 0.016}$ |
| FD-Dir-PP | $0.132 \pm 0.031$ | $0.106 \pm 0.004$ | $0.143 \pm 0.022$ | $0.375 \pm 0.007$ |
| Hawkes | $0.412 \pm 0.091$ | $0.158 \pm 0.005$ | $0.170 \pm 0.035$ | $0.507 \pm 0.003$ |
| RMTPP | $0.860 \pm 0.004$ | $0.257 \pm 0.005$ | $0.474 \pm 0.016$ | $0.721 \pm 0.001$ |
| F-LSTM | $0.277 \pm 0.118$ | $0.141 \pm 0.002$ | $0.209 \pm 0.023$ | - |
| F-RNN | $0.516 \pm 0.105$ | $0.146 \pm 0.004$ | $0.186 \pm 0.011$ | - |
| LSTM | $0.860 \pm 0.010$ | $0.251 \pm 0.008$ | $0.376 \pm 0.026$ | - |
| RNN | $0.841 \pm 0.016$ | $0.258 \pm 0.008$ | $0.439 \pm 0.017$ | - |

## H.2 Results

Tables 1 and 2, together with Fig. 16 show test results for *all* models on *all* datasets for Class accuracy and Time-Error.

Figure 16: Class accuracy (top) and Time-Error (bottom) comparison across datasets

## H.3 Time Prediction with Point Processes

The benefit of the point process framework is the ability to get the point estimate for the time $\hat{\tau}$ of the next event:

$$\hat{\tau} = \int_0^\infty t q(\tau) dt \tag{H.9}$$

where

$$q(\tau) = \lambda_0(\tau) \exp \left( - \int_0^\tau \lambda_0(s) ds \right) \tag{H.10}$$

The usual way to evaluate the quality of this prediction is using an MSE score. As we have already discussed in Sec. 5, this is not optimal for our use case. Nevertheless, we did preliminary experiments comparing our neural point process model **FD-Dir-PP** to others. We use **RMTPP** [5] since it achieves the best results. On Car Indicators dataset our model has mean MSE score of 0.4783 while RMTPP achieves 0.4736. At the same time FD-Dir-PP outperforms RMTPP on other tasks (see Sec. 5).