[Reviews · NeurIPS 2019]

Reviewer 1



This paper makes a compelling case for explicitly modelling a time varying distribution over the simplex for asynchronous events prediction. This in itself makes the paper a valuable contribution. The use of gaussian process to model the time dependency along with pseudo point from RNN is original, but not particularly well motivated. The authors even mention that an alternative of modeling the time evolution of distribution using RNN directly, which seems like the natural approach to avoid relying on pseudo points. However the two approaches merits and disadvantages are never compared. The paper is mostly well written and motivates the methodology well, but could do with more various topics such as the impact of UCE vs CE, how training looks in practice, or the choice of many pseudo points to use.

Reviewer 2



This paper introduces 2 methods to make predictions of asynchronous multi-class events. The methods pay special attention to giving confidence values to their predictions. The models model the changing distribution of outcomes given a time in the future, assuming that no other event has happened in the meantime. The first method (WGP-LN) uses an RNN to output a set of M control (pseudo) points to mode a Gaussian process. At first they describe learning a Gaussian process for each output class, but they discover that this leads to overconfidence at the control points. They then proceed to train the RNN to also output weights for the Gaussian process. Since there are a small number of control points (<10) the Gaussian process is fast to compute, and is also differentiable, thus enabling fast training with sgd. The second method uses the Dirichlet distribution, which is similar in practice, the RNN still outputs a set of Gaussian basis functions and weights. The calculation of loss for this method involves proposed closed-form expressions, which are then approximated with a second order series expansion. They also define a loss function (the Uncertainty cross-entropy loss) which is more suited to learn uncertainty on the categorical distribution. Also introduced it a point-process framework which can be used to predict the most likely time the next event is expected. The related work section seems a bit thin, but I am not an area expert. They provide many experimental results which show their method consistently out performing other methods. They generate toy examples, and show an ability to predict the evolution of next events, along with the confidence for those predictions. The appendix appears to cover the details of the methods and results. The supplementary materials also contains an excellent looking iPython notebook containing Tensorflow implementations of the methods and toy examples. I think the problem described (asynchronous, multi-class event prediction WITH attention being paid to the confidence of the prediction) is of great importance to the community, and these methods appear to be solid contributions. The code may be useful to many. Provided other reviewers pass the math, I think this is a good paper.

Reviewer 3



The authors target a very particular problem, that of predicting the uncertainty in predicting the type of event which is going to happen asynchronously in the future and where the probability of the event is dependent on the time. This is an important problem and is different from other settings in uncertainty prediction which have been explored elsewhere. The paper is very well written (besides some minor reorganization issues) and the illustrations are of high quality. The techniques described are sound and novel, and bringing them together is an important contribution. The experiments are described in adequate detail and the provided code is relatively easy to parse through and re-run to reproduce a subset of the results. However, there are two points which could improve the quality and clarity of the submission. The first is that the related work [6, 13] is mildly mischaracterized, and the cost of introducing sampling are not fully elucidated. Most notably, Neural Hawkes Process can successfully model multi-modal distributions of events, akin to FD-Dir-PP do it, and, hence, its characterization (e.g. in line 230) could be made better. Similarly, while it is true that RMTPP models type and time of the next event independently of each other (which precludes multi-modal event distributions) it does so in order to allow rapid training and efficient use of GPUs. This is a subtle difference, which leads to the second point of the true cost of the sampling step. Tensorflow, and GPU based training in general, works best when the entire training iteration happens on the GPU (i.e. no use of feed_dict with computed values). However, the sampling step, to the best of my knowledge, cannot be done on the GPU, and needs to happen on the CPU. The Neural Hawkes Process [13] too suffers from this, because they need to perform Monte-Carlo sampling in order to numerically evaluate integrals. This otherwise minor detail introduces a significant bottleneck in the training times. I believe an honest discussion of the pros and cons of the approach adopted would further embellish the paper and help put the contributions in context. Along the same line, it is unclear what the tradeoff of increasing/decreasing the number of samples 'M' is on the training time/quality and that pareto-optimal front would also be interesting to speculate/demonstrate. Given the overall contributions of the paper, I do not have any hesitation in recommending it for publication. Minor points: - Line 131: "a a" - Line 94: y_j^{(c)} is not defined without looking at the appendix. - Line 137: Claim about the model being fully differentiable is not justified until the loss function is provided. - Line 386: First denominator, the sum should go till C. - Line 444: Missing Figure reference. - Line 459: Comment in blue. ---------------------- Update after the author response: I have read the response and it has cleared up some misconceptions I had. A more nuanced treatment of the related work will also be appreciated.

[Author Response · NeurIPS 2019]

# Uncertainty on Asynchronous Event Prediction: Author Response

**Compound distribution.** We would like to emphasize that the uncertainty is not modeled through a compound distribution in our models. Indeed, the compound distribution would be $\text{Cat}(\bar{\boldsymbol{p}}_i(\tau))$ where $\bar{\boldsymbol{p}}_i(\tau) = \mathbb{E}_{\boldsymbol{p} \sim P_i(\theta(\tau))}[\boldsymbol{p}] = \int \boldsymbol{p} P_i(\theta(\tau))(\boldsymbol{p}) d\boldsymbol{p}$, and (see lines 167-169) the CE loss would only use this distribution. In contrast, the UCE does not use the compound distribution but considers the expected cross-entropy (note the order of $\int$ and $\log$).

**Intuition on UCE.** To give more intuition about the UCE loss, we propose the following example where we have two distributions on the simplex $P_i^{(1)}(\theta(\tau_i^*))$ and $P_i^{(2)}(\theta(\tau_i^*))$ such that $\bar{\boldsymbol{p}}_i(\tau_i^*) = \mathbb{E}_{\boldsymbol{p} \sim P_i^{(1)}(\theta(\tau_i^*))}[\boldsymbol{p}] = \mathbb{E}_{\boldsymbol{p} \sim P_i^{(2)}(\theta(\tau_i^*))}[\boldsymbol{p}]$. In this case the CE will be the same for both distributions, $\mathcal{L}_i^{(1)\,\text{CE}} = \mathcal{L}_i^{(2)\,\text{CE}}$. Now assume that all the probability mass is concentrated around the mean $\bar{\boldsymbol{p}}_i(\tau_i^*)$ for $P_i^{(1)}(\theta(\tau_i^*))$ but not for $P_i^{(2)}(\theta(\tau_i^*))$. Hence, $P_i^{(1)}(\theta(\tau_i^*))$ is very certain on the mean prediction. In contrast to CE, the UCE can distinguish the two distributions and especially $\mathcal{L}_i^{(1)\,\text{UCE}} < \mathcal{L}_i^{(2)\,\text{UCE}}$. Hence, an important property of the UCE is that the variance of the distribution on the simplex plays a substantial role in its value. In particular, high variance is penalized by the UCE which is particularly important during training. Indeed, the UCE will reduce the uncertainty for the categorical distributions predicted for the observed data. In combination with a prior value for the variance (which is done by the regularization term, lines 186-199), we keep the variance high for non-observed data while being more certain on the data we observed, as desired. Note that the regularization applied with CE would only set the variance of all (observed and non-observed) data/time points to the same prior. The CE would not reduce the variance on observed data and only adjust the mean prediction.

**Objective criteria for loss selection.** We propose the anomaly detection experiment with the distribution uncertainty (lines 305-322) as an objective criteria. The comparison of the different losses (CE, CE + reg, UCE + reg) for the FD-Dir model are shown in Fig. 1. The loss UCE + reg consistently improves the anomaly detection based on the distribution uncertainty. Furthermore, in the appendix, we proposed a visual representation of the benefit of UCE compared to CE on a simple classification task.

**Number of pseudo points.** In our initial hyperparameter search we tuned the number of points but for the final results we kept it fixed across datasets (see lines 485-486 in the supp. mat.). Figure 2 shows that changing the number of points does not significantly affect the accuracy (same for the other datasets). Additionally, Figure 5 in the paper shows that both models learn to give lower weights to unnecessary points, essentially discarding them if we have too many.

**Training time w.r.t. $M$.** If the size of the RNN's hidden state is $D$, and we have $M$ pseudo points, adding one more point leads to $D$ more parameters. In the case of GP, we have to take into account the increase in computation time due to the inverse. Since the number of points is always lower than $D$ and often $M < 10$, the increase is negligible. We found that the number of epochs until the early stopping is similar for different $M$. Therefore, neither the accuracy (see above) nor the training time are strongly affected when varying $M$.

Figure 1: Loss comparison

Figure 2: Number of pseudo points

**Sampling.** The Neural Hawkes Process [13] needs sampling to evaluate the integral and does so by passing time points through the RNN-based model which is expensive. In our case, sampling is (i) only required if we wish to use regularization or a point process version (note that obtaining the $M$ pseudo points does *not* require sampling), and (ii) very cheap. The reason is that the evolution of the distributions over time is represented by pseudo points, which after computing the RNN's hidden state are given. That is, for the Dirichlet model, sampling only requires to evaluate the Gaussian function; and for the GP model to evaluate the kernel function. The computation of the hidden state, the inverse of the covariance etc. can all be reused across multiple samples. We will add these discussions to the paper.

**Related Work.** We will extend the related work section based on your feedback. In particular, we will mention the ability of Neural Hawkes Process to model multi-modal distributions and the possibility of RMTPP to model decaying intensities (like many classic point processes, e.g. Cox, Hawkes).

[Meta-Review · NeurIPS 2019]

The paper introduces new techniques to model changes in categorical distributions over event types and the uncertainty in said distributions, where events happen asynchronously (not at pre-specified time instants). The goal is to predict the type of the next event conditioned on an observed history and a particular time gap, while correctly modeling the effect of time gaps on event-type predictions and their associated uncertainty in predictions. The reviewer scores were 5, 8, 8. All reviewers felt the problem setting and approaches were well motivated and the contributions were “important”, “practical”, “sensible”, etc. The reviewers appreciated the quality of the writing and the code submission with examples. R1 had some specific questions, which were largely addressed in the author feedback. The consensus is that this is a good paper and worthy of acceptance.